# Neutrophil Counts, Neutrophil-to-Lymphocyte Ratio, and Systemic Inflammatory Response Index (SIRI) Predict Mortality after Off-Pump Coronary Artery Bypass Surgery

**DOI:** 10.3390/cells11071124

**Published:** 2022-03-26

**Authors:** Tomasz Urbanowicz, Michał Michalak, Anna Olasińska-Wiśniewska, Michał Rodzki, Anna Witkowska, Aleksandra Gąsecka, Piotr Buczkowski, Bartłomiej Perek, Marek Jemielity

**Affiliations:** 1Cardiac Surgery and Transplantology Department, Poznan University of Medical Sciences, 61-848 Poznan, Poland; annaolasinska@ump.edu.pl (A.O.-W.); michal.rodzki@skpp.edu.pl (M.R.); anna.witkowska2@skpp.edu.pl (A.W.); piotr.buczkowski@skpp.edu.pl (P.B.); bperek@ump.edu.pl (B.P.); mjemielity@poczta.onet.pl (M.J.); 2Department of Computer Science and Statistics, Poznan University of Medical Sciences, 61-806 Poznan, Poland; michal@ump.edu.pl; 31st Chair and Department of Cardiology, Medical University of Warsaw, 02-091 Warsaw, Poland; gaseckaa@gmail.com

**Keywords:** OPCAB, NLR, SIRI, AISI, SII

## Abstract

Background: Several perioperative inflammatory markers are postulated to be significant factors for long-term survival after off-pump coronary artery bypass surgery (OPCAB). Hematological parameters, whether single or combined as indices, provide higher predictive values. Methods: The study group comprised 538 consecutive patients (125 (23%) females and 413 (77%) males) with a mean age of 65 ± 9 years, who underwent OPCAB with a mean follow-up time of 4.7 ± 1.7 years. This single-center retrospective analysis included perioperative inflammatory markers such as the neutrophil-to-lymphocyte ratio (NLR), systemic inflammatory response index (SIRI), aggregate index of systemic inflammation (AISI), and systemic inflammatory index (SII). Results: Multivariable analysis identified levels of neutrophils above 4.3 × 10^9^/L (HR 13.44, 95% CI 1.05–3.68, *p* = 0.037), values of SIRI above 5.4 (HR 0.29, 95% CI 0.09–0.92, *p* = 0.036) and values of NLR above 3.5 (HR 2.21, 95% CI 1.48–3.32, *p* < 0.001) as being significant predictors of long-term mortality. The multifactorial models revealed the possibility of strong prediction by combining preoperative factors (COPD, stroke, PAD, and preoperative PLR) and postoperative neutrophil counts (*p* = 0.0136) or NLR (*p* = 0.0136) or SIRI (*p* = 0.0136). Conclusions: Among the postoperative inflammatory indices, the levels of neutrophils, NLR, and SIRI are the most prominent markers for long-term survival after off-pump coronary artery bypass surgery, when combined with preoperative characteristics.

## 1. Introduction

Coronary atherosclerosis, combined with co-morbidities including obesity, diabetes, and arterial hypertension, as well as gender differences and psychosocial work stress factors, is still a major epidemiological challenge for public health services [1,2,3,4,5]. The origin and progression of atherosclerotic plaques are currently considered to be related to inflammatory process activation [6,7,8].

Complex coronary artery disease can be treated by either coronary percutaneous or surgical revascularization [9,10,11]. The periprocedural inflammatory overreaction is one of the possible factors that indicate a worse long-term prognosis [12,13,14,15,16].

The relationship between surgical intervention and cardiopulmonary bypass has been widely postulated [17,18,19,20]. Despite surgical challenges, the off-pump technique (off-pump coronary artery bypass surgery, OPCAB) can be performed safely by experienced surgeons and may rule out the risk of inflammatory activation that is secondary to CPB application [21]. Despite the elimination of cardiopulmonary application in the OPCAB technique, off-pump surgery still possesses an inflammatory burden that has a detrimental effect on the long-term prognosis [22].

Several inflammatory markers, including the neutrophil-to-lymphocyte ratio (NLR), platelet-to-lymphocyte ratio (PLR), and monocyte-to-lymphocyte ratio (MLR), have a documented value for predicting worse survival rates following surgical revascularization [12,13].

Numerous novel inflammatory markers have been described that involve the systemic inflammatory response index (SIRI—the quotient of neutrophils and monocytes, divided by lymphocyte count), the aggregate index of systemic inflammation (AISI—the quotient of neutrophils, monocytes, and platelets, divided by lymphocyte count), and the systemic inflammatory index (SII), composed of the quotient of neutrophils and platelets divided by lymphocyte counts [23]. These three indices have been presented as possible mortality prognostic factors in different cardiovascular and non-cardiovascular diseases [24,25,26,27,28]. These novel indices—the aggregate index of systemic inflammation (AISI), systemic inflammatory index (SII), and systemic inflammatory response index (SIRI)—involve the main compounds of previously well-known inflammatory markers, such as neutrophils, monocytes, lymphocytes, and platelets [29,30].

The current study aimed to assess the value of different inflammatory markers, including the more common neutrophil counts and NLR, as well as these novel ones—AISI, SII, and SIRI—for mortality prediction in consecutive patients with chronic coronary syndrome treated with surgical revascularization using the off-pump technique. Furthermore, we also aimed to design a multifactorial model for long-term mortality prediction, to avoid the selectivity of a limitation to a single parameter.

## 2. Materials and Methods

The study group comprised 538 consecutive patients (125 (23%) females and 413 (77%) males) with a mean age of 65 ± 9 years, who underwent an off-pump coronary artery bypass grafting (OPCAB) procedure between January 2015 and December 2018 in our hospital. The current research presents a single-center study, for which we conducted a retrospective analysis of patients referred for surgical revascularization due to complex chronic coronary syndrome. The patients requiring concomitant valve surgery and those referred for surgery because of acute coronary syndrome were excluded from the study. Additional exclusion criteria included inflammatory, autoimmune, oncological, or hematological proliferative diseases.

The researchers abided by the principles of good clinical practice and the Declaration of Helsinki, and the study was approved by the Local Ethics Committee of the Medical University of Poznan (approval number: 55/20 from 16/01/2020).

The co-morbidities for the study sample included arterial hypertension in 379 patients (71%), diabetes mellitus in 175 patients (33%), hypercholesterolemia in 298 patients (55%), chronic obstructive pulmonary disease (COPD) in 47 patients (9%), and chronic kidney disease in 32 patients (6%)—defined as a glomerular filtration rate (GFR) of ≤60 mL/min/1.63 m^2^, according to the Cockcroft–Gault equation.

We analyzed the demographic, clinical, and laboratory data; the following indices were calculated to assess the numbers of neutrophils, monocytes, and platelets applied in the NLR, SIRI, SII, and ASIS calculations, utilizing a routine hematology analyzer (Sysmex Europe GmbH, Norderstedt, Germany).

Furthermore, we conducted echocardiography for each patient before the surgery, during hospitalization, and at discharge. Data concerning long-term mortality were collected from the outpatient clinic and the Polish National Health Service database.

### Statistics Analysis

We presented the continuous variables as a mean ± standard deviation (SD) or median with an interquartile range; we conducted the analysis using an unpaired Student’s *t*-test or Mann–Whitney U test since the data did not follow a normal distribution. We presented the categorical variables as frequencies and percentages and analyzed them using a test for proportions. We deployed receiver-operating characteristics (ROC) to determine the cut-off values of the analyzed predictors, to discriminate between individuals enrolled in the study, grouping them into those with and without a mortality endpoint. We also used the log-rank test to check the significance of the survival curves, while using Cox’s proportional hazards model to analyze the long-term mortality predictors. We performed univariate and multivariate analyses (stepwise, backward selection procedure). Furthermore, we transformed the continuous parameters into binary ones (via ROC analysis) to unify the data types. We implemented the hazard ratios (HR) and 95% confidence intervals (95% CI) to interpret and infer from the results.

## 3. Results

### 3.1. Clinical Results

During the 4.7 ± 1.7 years of follow-up, fifty-one patients died, irrespective of the cause of death (all-cause mortality of 10%). The survivors and non-survivors did not differ significantly regarding gender (*p* = 0.589) and age (*p* = 0.161) (Table 1). In the pre-procedural transthoracic echocardiography (TTE) test, the median left ventricle end-diastolic diameter was not significantly larger (*p* = 0.056) in the non-survivor group than in survivors, with median values of 50 mm (45–54 mm) and 47 mm (44–52 mm), respectively. There was also a significant difference (*p* = 0.032) in the preoperative left ventricle ejection fraction (LVEF), with median values of 55% (50–60) vs. 50% (45–60) in survivors and non-survivor groups, respectively.

The main indication for surgery was three-vessel disease in 188 (35%) patients, followed by left main stem stenosis in 178 (33%) patients, and two-vessel disease in 172 (32%) patients. The mean surgery (skin-to-skin) time was 2.3 ± 0.5 h, and the mean number of performed anastomoses was 2.25 ± 0.2. None of the surgeries were performed as repeat surgery. There were no intra-operative deaths. The 30-day mortality rate was 1.2% (seven patients). Excessive bleeding episodes requiring re-thoracotomy occurred in 18 patients (3%); the median time of intensive care unit (ICU) stay was 27 h (17–35 h) for the presented group. The demographical and clinical characteristics are presented in Table 1.

Perioperative characteristics, including surgical parameters, laboratory test results, and echocardiographic data, were analyzed and revealed significant differences among groups in terms of laboratory inflammatory indexes—SIRI (*p* = 0.012), SII (*p* < 0.001), AISI (*p* < 0.001). All the significant results are presented in Table 2.

Neither the mean number of performed grafts (2.25 ± 0.3 vs. 2.2 ± 0.2 (*p* = 0.798)), nor the postoperative maximum values of serum Troponin-I (1.5 (0.8–3.5) ng/mL vs. 1.8 (0.6–5.4) ng/mL (*p* = 0.578)), nor hospitalization length (11 ± 4 days vs. 10 ± 3 days (*p* = 0.821)) differed between the survivor and non-survivor groups.

Postoperative laboratory characteristics revealed significant differences between groups in terms of neutrophils (*p* = 0.003), platelets (*p* = 0.009), NLR (*p* < 0.001), PLR (*p* < 0.001), SIRI (*p* = 0.012), SII (*p* < 0.001), and AISI (*p* < 0.001). 

### 3.2. Receiver Operator Characteristics (ROC) Analysis

We focused on novel inflammatory markers for long-term mortality prediction after off-pump surgery. ROC analysis revealed significant results for postoperative values of SIRI (AUC = 0.616, *p* = 0.008), yielding a sensitivity of 52.27% and specificity of 69.92%, with a cut-off value of 5.4; SII (AUC = 0.669, *p* = 0.001) yielded a sensitivity of 60.00% and specificity of 70.11%, with a cut-off value above 953; AISI (AUC = 0.659, *p* = 0.0001) yielded a sensitivity of 70% and a specificity of 56.70%, with a cut-off value above 663. The results are presented in Figure 1A–C.

### 3.3. Univariable Analysis

We performed the univariate Cox regression analysis, in which co-morbidities (COPD, stroke, peripheral artery disease (PAD)), preoperative (PLR and serum creatinine), and postoperative parameters, including the inflammatory indexes (SIRI, SII, AISI), followed by the echocardiographic left ventricle ejection fraction were marked as significant risk factors for long-term survival. Co-existing diseases, such as COPD (HR = 2.51, 95% CI 1.29–4.88, *p* = 0.007), stroke (HR = 4.80, 95% CI 2.53–9.10, *p* < 0.001), and PAD (HR = 2.96, 95% CI 1.62–5.39, *p* < 0.001) were statistically significant. Among the preoperative laboratory parameters, PLR (HR = 1.00, 95% CI 1.00–1.01, *p* = 0.032) and serum creatinine level (HR = 2.59, 95% CI 1.04–6.51, *p* = 0.042) appeared to be statistically significant. The postoperative parameters are presented in Table 3; among others, a SIRI level above 5.4 (HR = 2.05, 95% CI 1.10–3.83, *p* = 0.025), an SII level above 953 (HR = 3.26, 95% CI 1.81–5.88, *p* < 0.001), and an AISI level above 663 (HR = 2.82, 95% CI 1.48–5.39, *p* = 0.002) were significant for long-term mortality.

### 3.4. Multivariable Analysis

Parameters that were estimated as significant in the univariable analysis were then verified in a multivariable analysis. Co-morbidities such as COPD (HR = 10.58, 95% CI 2.42–46.36, *p* = 0.002), stroke (HR = 19.25, 95% CI 5.54–66.94, *p* < 0.001), and peripheral artery disease (HR = 3.78, 95% CI 1.28–11.15, *p* = 0.016) were found to represent significant risk factors as presented in Table 4. The preoperative PLR (HR = 0.98, 95% CI 0.96–0.99, *p* = 0.001) was the only preoperative laboratory parameter influencing long-term survival. The postoperative hemoglobin levels (HR = 3.27, 95% CI 1.09–2.79, *p* = 0.018), creatinine levels (HR = 1.2, 95% CI 1.01–10.4, *p* = 0.003), neutrophils at 4.3 × 10^9^/L above the cut-off value (HR = 13.44, 95% CI 1.05–3.68, *p* = 0.037), SIRI at 5.4 above the cut-off value (HR = 0.29, 95% CI 0.09–0.92, *p* = 0.036), and NLR at 3.5 above the cut-off value (HR = 2.21, 95% CI 1.48–3.32, *p* < 0.001) presented significant values for long-term mortality prediction.

### 3.5. Receiver Operator Curve for Postoperative Inflammatory Markers Revealed in the Multivariable Analysis

We compared three postoperative inflammatory indices related to neutrophils, which revealed significant values for long-term mortality prediction in the multivariable analysis. 

The receiver operator characteristics curve (ROC) was established for neutrophil counts (AUC = 0.628, *p* = 0.001), yielding a sensitivity of 86% and a specificity of 39.1%, with a cut-off value of 4.3 (Figure 2A); for the neutrophil to lymphocyte ratio (AUC = 0.643, *p* = 0.001), this yielded a sensitivity of 50% and a specificity of 76.9%, with a cut-off value of above 3.5 (Figure 2B); the systemic index of SIRI (AUC = 0.616, *p* = 0.008) yielded a sensitivity of 52.27% and a specificity of 69.92%, with a cut-off value of above 5.4 (Figure 2C).

### 3.6. Receiver Operator Curve for Multifactor Models, including Factors Presented in Multivariable Analysis (Preoperative Factors and Postoperative Inflammatory Markers)

After off-pump surgery, all three inflammatory indices presented a mildly significant predictive value for long-term mortality risk prediction. We performed further analyses, including an analysis of preoperative factors. ROC analysis of long-term mortality prediction, including multifactor score, was performed. We compared the ROC curve as the combination of inflammatory markers (neutrophil counts, NLR, or SIRI) with preoperative factors (stroke, peripheral artery disease, COPD, and PLR). The ROC analysis for neutrophils with a cut-off value of 4.3, combined with preoperative factors, is presented in Figure 3A (AUC = 0.787, *p* < 0.001), yielding a sensitivity of 78.82% and a specificity of 64.49%. The ROC analysis for NLR with a cut-off value above 3.5. Together with preoperative factors, this is presented in Figure 3B (AUC = 0.767, *p* < 0.001), yielding a sensitivity of 61.70% and a specificity of 81.86%. The ROC analysis for the systemic index, SIRI, with a cut-off value of above 5.4 and preoperative factors are presented in Figure 3C (AUC = 0.787, *p* < 0.001), yielding a sensitivity of 75.61% and a specificity of 67.51%.

### 3.7. Multifactorial Models Analysis

The pairwise analysis presented similar results, independent of the inflammatory indices and in addition to the neutrophil counts, as presented in Table 5.

Inclusion of the preoperative factors (COPD, stroke, PAD, and preoperative PLR) to any of the three postoperative inflammatory indices, irrespective of neutrophil counts (AUC = 0.787), NLR (AUC = 0.767), or SIRI (AUC = 0.783), presented a comparable quality of constructed models, as presented in Figure 4.

## 4. Discussion

To the best of our knowledge, this is the first study presenting the values of different inflammatory indices obtained from peripheral blood count and their comparisons for the purposes of mortality prediction after off-pump coronary bypass surgery. There is a general opinion on the significant value of combined indices, which comprise neutrophil counts from the whole blood analysis and other components, including lymphocytes, platelets, and monocytes. Moreover, demographic and clinical factors seem to present a similar significance.

Neutrophils are defined as short-lived and unrefined phagocytes whose activation is triggered by either bacterial infection or immune activation [31]. Their missions of releasing vast numbers of proteolytic enzymes and reactive oxygen species are responsible for the recruitment and activation of monocytes, macrophages, and dendritic cell subsets [32]. The neutrophil’s life span is dependent on growth factors and cytokine modulation. Neutrophils undergoing apoptosis ameliorate inflammatory processes; conversely, when they become necrotic, they perpetuate inflammation. The latter processes are claimed to stimulate atherosclerotic lesions [33]. Neutrophil granulocyte markers were detected in human carotid arteries in atherosclerotic specimens, supporting their significance in plaque formation [34]. Consistent with previous reports on the impact of activated neutrophils on atherosclerosis plaque formation, our study revealed the relationship between perioperative inflammatory activation and its possible influence on an increased risk regarding long-term mortality. 

We evaluated the more commonly used neutrophil counts and NLR, in addition to multifactorial indices. We used univariate modeling to show the significance of neutrophil counts, NLR, SIRI, SII, and AISI. In the next step, we performed a combination of indices and clinical parameters that were shown to be predictive in multivariate analysis. We found that neutrophil counts and their derivative, NLR, have the highest predictive value for estimating mortality, following pairwise analysis. Therefore, we suggest that the neutrophil count and NLR are the most accurate factors and should be predominantly considered for assessing long-term prognoses. Other indices were significant in our univariate analysis. However, we must point out that all the indices mentioned above comprise neutrophil count or NLR; therefore, this observation further confirms the impact of neutrophils.

The main finding of our study encompasses the dominant significance of neutrophil counts on long-term mortality prediction, either alone or in composition, presented as NLR or SIRI. The combination of preoperative factors with postoperative neutrophil counts (above 4.3 × 10^9^/L) or NLR (above 3.5) showed a significant predictive value, with a sensitivity of 78.82% vs. 78.72% and a specificity of 65.49% vs. 64.49%, respectively. Moreover, the pairwise analysis, preceded by multivariable comparison, indicated the significant role of those postoperative inflammatory factors with preoperative ones, to validate a more robust predictive model. 

We focused on the perioperative inflammatory response to surgical intervention as a predictive factor due to the reported significance of inflammatory reactions for long-term survival [35,36]. The predictive scores that apply in clinical practice include the EuroSCORE II or STS score; these are dedicated to assessing perioperative mortality and the risk of complications [37,38,39,40,41]. 

The research on the inflammatory origin of atherosclerosis enables a better understanding of the pathophysiological background of the disease [42,43]. Several reports proved a strict relationship between the progression of coronary disease, inflammatory processes, and mortality [44,45,46]. The recruitment of a particular lineage of immune cells, such as neutrophils, monocytes, and lymphocytes, occurs during revascularization and has a detrimental effect on long-term prognosis [47,48,49,50]. The simple counts of neutrophils and monocyte-to-lymphocyte ratio were proposed as prognostic markers [51,52,53,54,55]. The perioperative inflammatory reactions, presented as the ratios mentioned above, were associated with long-term surgical revascularization results [56,57,58]. Novel markers, such as AISI, SII, and SIRI, were proposed to augment the propensity values of hematological indices in long-term prognosis [59,60,61].

We compared the utility of novel inflammatory systemic indexes in long-term predictions regarding off-pump surgery. Yamamoto et al. showed dynamic changes in oxidative stress components in early reperfusion after surgery, representing an inflammatory response to the procedure [62]. Interestingly, in our analysis, inflammatory markers related to neutrophils appeared to present the highest significance for long-term mortality prediction. The neutrophil counts, neutrophil to lymphocyte ratio, and the systemic inflammatory response were revealed to be significant in multivariable analysis. The comparison of those three indices in our study validates the claim that neutrophil counts and the neutrophil to lymphocyte ratio are the predictive factors.

The systemic inflammatory index (SII) is an inflammatory index that integrates three types of cells that are involved in immune response, including neutrophils, lymphocytes, and platelets. Previously, we described the particular features of all mentioned cells and the significance of their indices (NLR, PLR) to immune response in patients with coronary artery disease [61]. SII is an established prognostic marker of long-term prognosis in coronary revascularization [12], non-cardiac surgery, or neoplasms [62,63]. Platelets, in turn, are also involved in the inflammatory response by facilitating the recruitment of other inflammatory cells and the release of inflammatory mediators [64]. Therefore, SII should better reflect the inflammatory status compared to the three types of cells when assessed separately. Researchers have already described it as a prognostic marker in appendicitis [65], coronary artery disease [66] and neoplasms [67,68]. Patients with a high level of SII had a significantly higher mortality rate than those with a low SII [69].

Interestingly, patients with different dietary patterns showed differences in SII and NLR values in the study by Szymanska et al. (2021) [70]. Those with a lower ratio of omega-6 to omega-3 fatty acids had lower SII and NLR values, which resulted from an anti-inflammatory diet. Adali and collaborators underlined the significance of coronary artery disease pharmacotherapy [71]. Patients treated with ticagrelor had lower SII, NLR, and PLR values than clopidogrel-treated patients. In our study, an SII above 953 was significant for long-term mortality predictions in the univariate analysis. The most plausible explanation is that all components of SII, neutrophils, lymphocytes, and platelets play roles in atherosclerosis initiation, modulation, and aggravation. Thus, higher cell counts may indicate an ongoing disease and augment the phenomenon. Therefore, higher SII values directly show the risk of disease progression and indicate a potentially worse prognosis. SII use, however, appeared to be limited and was not confirmed in our multivariable analysis.

The AISI—aggregate index of systemic inflammation—is similar to the SII, but in addition to neutrophils, lymphocytes, and platelets, it also includes the monocyte count. Monocytes and neutrophils are part of innate immunity and produce proinflammatory cytokines, chemokines, enzymes, and reactive oxidative species. After activation, monocytes may transform into foam cells, destabilizing the atherosclerotic plaque and promoting dysfunctional and atherogenic lipoproteins [56]. Although AISI represents several inflammatory cells as ingredients and should provide more precise prognostic value, it is rarely used or described in the literature. Most recently, Zinellu et al. (2021) underlined its prognostic significance to predict poor outcomes in idiopathic pulmonary fibrosis [72]. Furthermore, Hamad and coworkers (2021) also studied its importance in patients suffering from COVID-19 [23]. Although it was significant in univariate analysis, like SII, it did not convey a sufficient prognostic value in the multivariate analysis. 

The systemic inflammatory response index (SIRI) is the immune system’s reaction against infection and invasive pathogens [73]; the name is given to an inflammatory index describing immunological defenses that encompass neutrophils, monocytes, and lymphocytes [74,75,76]. Surgical intervention may serve as an initiative non-immunological trigger [77]. The main issue of the study was the relationship between inflammation activity secondary to the surgery and long-term mortality. Even though inflammatory activation secondary to the off-pump procedure is limited, these reactions still play a crucial role as long-term mortality predictors [78]. We found that SIRI was a significant predictor [79], especially when we integrated data from preoperative factors.

Neutrophils regulate the repair processes in response to injury, inflammatory reactions, and neoplasms. Neutrophils may also contribute to adaptive changes in developing specific adaptive immune responses and may even trigger tissue damage when inappropriate inflammation reactions are activated [80,81]. Both monocytes and neutrophils are attracted to inflammatory sites via cytokines. Mature monocytes are released into the circulation and are recruited for inflammation control and tissue repair [82]. They represent immune cells involved in emerging human inflammatory diseases such as atherosclerosis [83]; their role in the early formation and maturation of atherosclerotic plaques is crucial [84]. The monocytes characterized by the expression of proinflammatory genes and phenotypes were found in patients with coronary artery disease [85]. Lymphocytes are recruited to the sites of inflammation and contribute to chronic inflammatory processes [86]. The lymphocytes’ memory of past stimuli triggers the production of effector cytokines, proliferation, and performance effector functions [87]. The inflammatory markers, including the earlier cells, appear to accurately distinguish the prognostic factors of hematologic response to immune system activation. The proposed SIRI marker is less widely explored than the more commonly used NLR [88,89,90,91,92].

The results of our study identify patients who are more prone to SIRI component activation secondary to a surgical trigger. This group is characterized by poorer long-term prognoses. We believe that a single stimulus, such as surgery, may reveal an individual propensity for inflammatory component activation that may lead to atherosclerosis progression. Regarding previous studies [93], neutrophils may initiate plaque formation via multiple roles, but they are also involved in their destabilization [94]. The circulating monocytes in patients with complex coronary artery disease have been postulated to present an increased capacity for cytokine production [94]. Monocytes and macrophages may change their characteristics, secondary to pro-atherogenic stimuli [95], and participate in plaque formation [96]. After endothelial damage, the chemoattractant protein-1 (MCP-1) derived from macrocytes will initiate monocyte migration and secondary foam cell formation, as monocytes are believed to represent one of the initial steps of atherogenesis [97]. The progression of plaques is also related to lymphocyte activation, including B2 and T lymphocytes [98]. Atherosclerotic plaque destabilization, related to activation of neutrophils [99,100], monocytes [101] and lymphocytes [102,103] has also been postulated.

We performed a comparison of these three parameters (neutrophil counts, NLR, and SIRI) to estimate their clinical value for long-term prognosis following off-pump coronary artery bypass grafting. Our multivariate analysis revealed the significance of neutrophil counts, NLR, and SIRI for long-term prognosis. The postoperative inflammatory parameters even possessed a stronger predictive effect when combined with preoperative factors. The analysis performed for this study indicates the two significant components influencing outcomes, represented by co-morbidities and perioperative inflammatory activation. Perioperative inflammatory activation is patient-dependent and possesses a predictive value [104,105,106], although this is only after compilation with preoperative factors. 

We want to emphasize that among the preoperative factors, apart from the concomitant diseases (co-morbidities), the platelet-to-lymphocyte ratio (PLR) was also marked as being significant in the multivariable analysis. The PLR reflects an increased level of inflammation and thrombosis, as presented in previous reports [107,108,109]. This preoperative factor may indicate that the individual propensity for inflammatory system activation should be considered before operating.

## 5. Conclusions

Among the postoperative inflammatory indices, and when combined with preoperative characteristics, the neutrophils, NLR, and SIRI represent the most prominent predictors for long-term survival after off-pump coronary artery bypass surgery. The multivariable analysis established that neutrophils above 4.3, a SIRI above 5.4, and an NLR above 3.5 were significant long-term outcome predictors. The predictive values of the ROC analysis achieved significance as multifactor models were constructed by compiling preoperative co-morbidities (stroke, COPD, PAD) and PLR with postoperative inflammatory markers.

## Figures and Tables

**Figure 1 cells-11-01124-f001:**
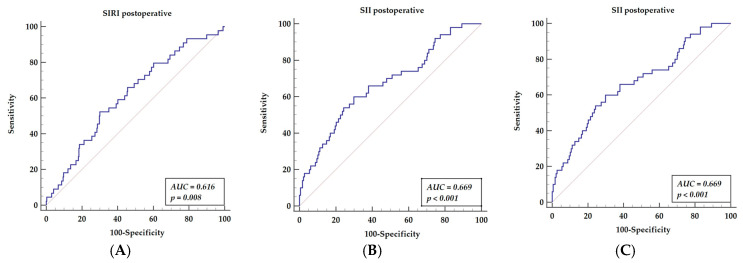
Receiver operator characteristics curves for postoperative SIRI (**A**), postoperative SII (**B**), and postoperative AISI (**C**). Abbreviations: AISI—aggregate index of systemic inflammation, AUC—area under the curve, SII—systemic inflammatory index, SIRI—systemic inflammatory response index.

**Figure 2 cells-11-01124-f002:**
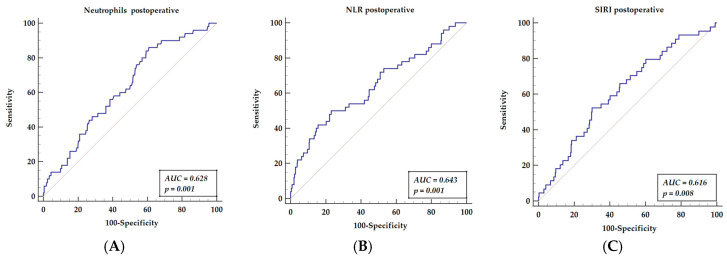
Receiver operator characteristics curves of three hematologic indices related to neutrophils-neutrophil counts (**A**), NLR (**B**), SIRI (**C**) that are significant for mortality prediction in multivariable analysis. Abbreviations: NLR—neutrophil to lymphocyte ratio, SIRI—systemic inflammatory response index.

**Figure 3 cells-11-01124-f003:**
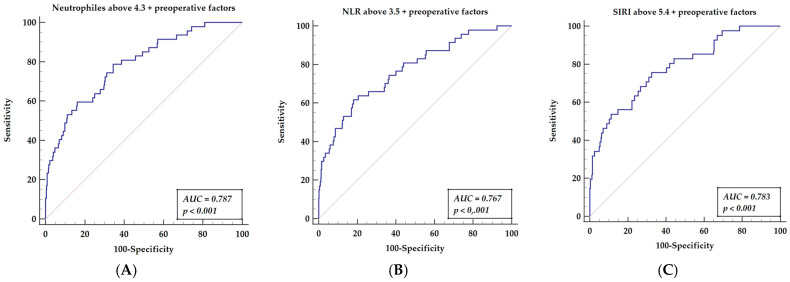
Receiver operator characteristics curves of three hematologic indices related to neutrophils (neutrophil counts (**A**), NLR (**B**), SIRI (**C**)), paired with preoperative factors that are significant for mortality prediction in multivariable analysis. Abbreviations: NLR—neutrophil-to-lymphocyte ratio, SIRI—systemic inflammatory response index. Preoperative factors = stroke, peripheral artery disease, COPD, and PLR.

**Figure 4 cells-11-01124-f004:**
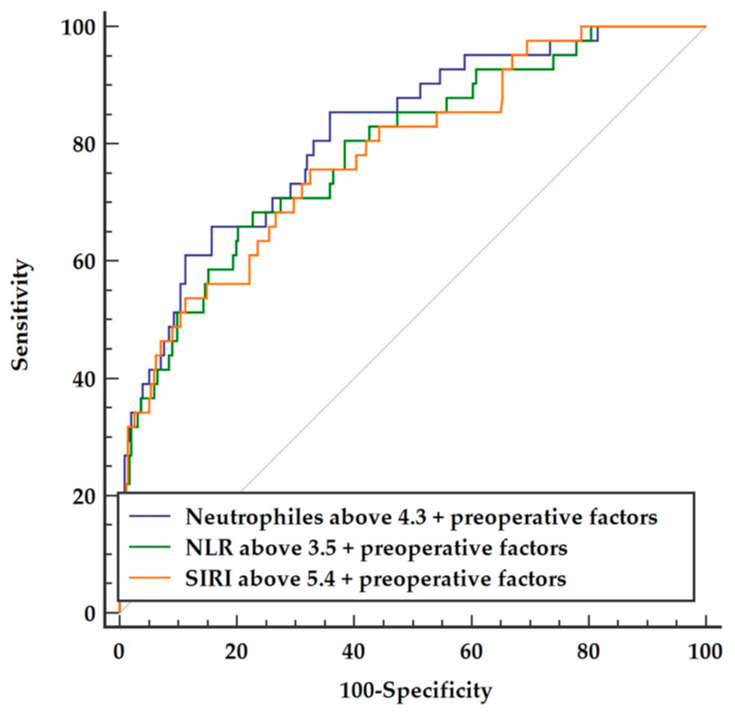
Comparison of three multifactor models including preoperative parameters and postoperative neutrophils vs. NLR vs. SIRI, respectively. Abbreviations: NLR—neutrophil-to-lymphocyte ratio, SIRI—systemic inflammatory response index.

**Table 1 cells-11-01124-t001:** Patient characteristics (demographical, clinical, and laboratory results).

	Survivors No. = 487 (90%)	Non-Survivors No. = 51 (10%)	*p*
Demographical data			
Sex (M/F)	371 (77%)/116 (23%)	42 (82%)/9 (18%)	0.589
Age (years)	64 (60–71)	67 (62–72)	0.161
Co-morbidities			
Arterial hypertension (*n* (%))	379 ((71%)	40 (78%)	0.109
Diabetes mellitus (*n* (%))	175 (33%)	16 (31%)	0.676
Hypercholesterolemia (*n* (%))	298 (55%)	29 (57%)	0.173
COPD (*n* (%))	47 (9%)	12 (24%)	<0.001 *
PAD (*n* (%))	80 (15%)	16 (31%)	<0.001 *
Kidney failure (*n* (%))	29 (6%)	3 (6%)	0.768
Laboratory tests:			
WBC × 10^9^/L (median (Q1–Q3))	7.8 (6.4–9.3)	7.5 (6.4–8.9)	0.388
Lymphocytes × 10^9^/L (median (Q1–Q3))	1.8 (1.4–2.2)	1.7 (1.3–2.0)	0.092
Neutrophils × 10^9^/L (median (Q1–Q3))	5 (4–6.3)	5.1 (4.2–6.1)	0.886
NLR (median (Q1–Q3))	2.8 (2–3.7)	2.8 (2.1–4.0)	0.235
Hb × 10^9^/L (median (Q1–Q3))	8.7 (8.2–9.2)	8.6 (7.9–9.3)	0.658
Platelets × 10^3^/μL (median (Q1–Q3))	225 (190–267)	230 (202–261)	0.456
Monocytes × 10^9^/L (median (Q1–Q3))	0.5 (0.4–0.6)	0.5 (0.3–0.6)	0.877
MLR (median (Q1–Q3))	0.3 (0.2–0.4)	0.3 (0.2–0.4)	0.113
MCHC (mmol/L) (median (Q1–Q3))	21.3 (20.8–21.7)	21 (20.6–21.1)	0.037
PLR (median (Q1–Q3))	125 (98–163)	140 (114–167)	0.027 *
Troponin I (ng/mL) (median (Q1–Q3))	0.01 (0.01–0.02)	0.02 (0.01–0.03)	0.13
Creatinine (mg/dL) (median (Q1–Q3))	85 (72–102)	99 (67–132)	0.044 *
SIRI (median (Q1–Q3)	1.3 (0.8–1.9)	1.3 (0.9–2.1)	0.261
SII (median (Q1–Q3))	618 (424–903)	668 (445–982)	0.174
AISI (median (Q1–Q3))	273 (172–440)	308 (185–489)	0.199

Abbreviations: AISI—aggregate index of systemic inflammation, COPD—chronic obstructive pulmonary disease, Hb—hemoglobin, LV—left ventricle, LVEF—left ventricle ejection fraction, MCHC—mean corpuscular hemoglobin concentration, MLR—monocyte-to-lymphocyte ratio, NLR—neutrophil-to-lymphocyte ratio, PAD—peripheral artery disease, PLR—platelets-to-lymphocyte ratio, SII—systemic inflammatory index, SIRI—systemic inflammatory response index, WBC—white blood cells. * Statistically significant difference. Continuous variables are expressed as the medians with the lower and the upper quartile, whereas categorical variables are expressed as (*n*) with a percentage (%).

**Table 2 cells-11-01124-t002:** Significant differences in postoperative laboratory characteristics between survivors and non-survivors.

	Survivors No. = 487	Non-Survivors No. = 51	*p*
Neutrophils × 10^9^/L (median (Q1–Q3))	4.9 (3.7–6.4)	5.7 (4.7–7.4)	0.003
NLR (median (Q1–Q3))	2.5 (1.8–3.4)	3.4 (2.3–5.6)	<0.001
Platelets × 10^3^/ μL (median (Q1–Q3))	274 (227–338)	321 (243–409)	0.009
PLR (median (Q1–Q3))	147 (227–338)	171 (140–237)	<0.001
SIRI (median (Q1–Q3))	4.1 (2.6–6.2)	5.5 (3.6–7.5)	0.012
SII (median (Q1–Q3))	699 (483–1053)	1074 (565–1590)	<0.001
AISI (median (Q1–Q3))	607 (370–1019)	989 (599–1604)	<0.001

Abbreviations: AISI—aggregate inflammatory response index, NLR—neutrophil to lymphocyte ratio, PLR—platelets to lymphocyte ratio, SII—systemic inflammatory index, SIRI—systemic inflammatory response index. Laboratory parameters were performed at the time of admission for surgery. Continuous variables are expressed as the medians with the lower and the upper quartile.

**Table 3 cells-11-01124-t003:** Cox regression univariable analysis for long-term survival.

Parameter	HR	95% CI	*p*-Value
Demographical and clinical:			
COPD	2.51	1.29–4.88	0.007
Stroke	4.8	2.53–9.10	<0.001
PAD	2.96	1.62–5.39	<0.001
Preoperative parameters:			
PLR	1	1.00–1.01	0.032
Creatinine	2.59	1.04–6.51	0.042
Postoperative parameters:			
Neutrophils	1.12	1.07–1.17	<0.001
Neutrophils > 4.3 × 10^9^/L	3.68	1.56–8.68	0.003
NLR	1.16	1.10–1.22	<0.001
NLR > 3.5	2.74	1.54–4.88	0.001
Platelets	1.05	1.00–1.01	0.002
PLR	1.01	1.00–1.01	0.001
SIRI > 5.4	2.05	1.10–3.83	0.025
SII	1	1.00–1.00	<0.001
SII > 953	3.26	1.81–5.88	<0.001
AISI	1	1.00–1.00	<0.001
AISI > 663	2.82	1.48–5.39	0.002
MLR	2.2	1.08–4.49	0.03
Echocardiographic:			
LVEF	0.928	0.90–0.95	<0.001
LVEF below 45%	4.41	2.43–8.03	<0.001

Abbreviations: AISI—aggregate index of systemic inflammation, COPD—chronic obstructive pulmonary disease, Hb—hemoglobin, LV—left ventricle, LVEF—left ventricle ejection fraction, MCHC—mean corpuscular hemoglobin concentration, MLR—monocyte to lymphocyte ratio, NLR—neutrophil to lymphocyte ratio, PAD—peripheral artery disease, PLR—platelet to lymphocyte ratio, SII—systemic inflammatory index, SIRI—systemic inflammatory response index. Preoperative laboratory parameters were performed at the time of admission for surgery, while postoperative laboratory parameters were performed 24 h after surgery.

**Table 4 cells-11-01124-t004:** Multivariable Cox regression model results.

Parameter	HR	95% CI	*p*-Value
Demographical and clinical:			
COPD	10.58	2.42–46.36	0.002
Stroke	19.25	5.54–66.94	<0.001
PAD	3.78	1.28–11.15	0.016
Laboratory parameters:			
preoperative PLR	0.98	0.96–0.99	0.001
postoperative Hb	3.27	1.09–2.79	0.018
Neutrophils > 4.3 × 10^9^/L	13.44	1.05–3.68	0.037
postoperative SIRI > 5.4	0.29	0.09–0.92	0.036
postoperative NLR > 3.5	2.21	1.48–3.32	<0.001
postoperative creatinine	1.02	1.01–10.4	0.003

Abbreviations: COPD—chronic obstructive pulmonary disease, Hb—hemoglobin, NLR—neutrophil to lymphocyte ratio, PAD—peripheral artery disease, PLR—platelet to lymphocyte ratio, SIRI—systemic inflammatory response index. Preoperative laboratory parameters were performed on admission for surgery, postoperative laboratory parameters were performed 24 h after surgery. Statistics were performed using a multivariable proportional hazard Cox regression model.

**Table 5 cells-11-01124-t005:** Results of different compositions of multifactorial models affecting long-term survival.

Variable	AUC	SE	95% CI	Sensitivity (%)	Specificity (%)
1. Neutrophils > 4.3 + preoperative factors	0.787	0.0355	0.748 to 0.822	78.72	65.49
2. NLR > 3.5 + preoperative factors	0.767	0.0388	0.728 to 0.804	61.7	81.86
3. SIRI > 5.4 + preoperative factors	0.783	0.0396	0.739 to 0.823	75.61	67.51

Abbreviations: AUC—area under the curve, CI—confidence interval, NLR—neutrophil-to-lymphocyte ratio, SE—standard error, SIRI—systemic inflammatory response index.

## Data Availability

All data will be available from the correspondence e-mail address for three years following the publication upon reasonable request.

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
