# Peer review of "Neutrophil Counts, Neutrophil-to-Lymphocyte Ratio, and Systemic Inflammatory Response Index (SIRI) Predict Mortality after Off-Pump Coronary Artery Bypass Surgery"

_cells, 2022, doi:10.3390/cells11071124_

Round 1

Reviewer 1 Report

This is a retrospective study looking into the association between several indices of inflammation, as defined by perioperative cells counts, and long-term mortality following off-pump cardiac surgery. The authors have used data from a cohort of 538 consecutive patients undergoing elective off-pump coronary artery bypass surgery. The results of the study demonstrate a statistically significant predictive value of a number of indices for long-term mortality.

Major comments:

Overall, the pathophysiological logic of the association under study is not entirely clear to me. I understand the association between coronary artery disease, inflammatory response and risk of plaque rupture. This is mainly relevant in the context of non-cardiac interventions and acute complications in the perioperative phase. However, the logic in the context of this study – cardiac surgery, which treats coronary artery disease – and with respect to the outcomes studied – long-term instead of acute complications – does not appeal to me. Moreover, the clinical relevance and implications of the association found in this study will be hard to explain, as the relation between the acute perioperative response and the long-term state of the immune system is not being studied. With the data available, the only way to potentially do is would be to look at the association between preoperative and postoperative cell counts.

Patient selection: Why were patients presenting with acute coronary syndrome excluded? How long after the index surgery were patients follow-up for?

It is unclear in the methods section when in the perioperative period the cell count values were obtained – please provide more detail here. This is of key importance since the character of the acute perioperative response will vary over time. It is particularly confusing as table 1 contains preoperative cell count values, while further in the analysis ‘postoperative’ values are used.

On several occasions in the methods actual results are mentioned. Please confine these to the results section.

The statistical analysis section is very unstructured – the order that the various steps are described in is not logical, which would make it impossible to repeat the analyses if data were provided. Also, transforming continuous variables into binary variables not only reduces statistical power significantly, but it also does not make a lot of sense in the context of the research question under study.

The three indices used are likely not independent, and it is therefore to be expected that there will be significant overlap between the various models presented. Correction for multiple testing should therefore likely be applied here.

It would be very useful if some more information were provided on residual confounding in this model. Given the very high HRs presented, it is hard to believe that there are no important either residual confounding or interactions that have not been accounted for in the modelling.

The way Table 2 is presented is misleading – displaying only significant results (of an uncorrected analysis) is not useful at all.

This manuscript is written in very poor English – both grammatically, as well as textual and typographic errors. Also, in various spot incorrect language is used (eg, L44: ‘cardiopulmonary bypass’ is incorrect. CA stands for Coronary Artery). It would need a significant revision before it can be considered again.

Author Response

Dear Reviewer 1,

I would like to thank you on behalf of all co-authors for your valuable suggestions.

I’m grateful you found the time and engagement to help us improve our manuscript.

We followed your suggestions as presented below.

          Kind regards

Tomasz Urbanowicz et. al.

Major comments:

Overall, the pathophysiological logic of the association under study is not entirely clear to me. I understand the association between coronary artery disease, inflammatory response, and risk of plaque rupture. This is mainly relevant in the context of non-cardiac interventions and acute complications in the perioperative phase. However, the logic in the context of this study – cardiac surgery, which treats coronary artery disease – and with respect to the outcomes studied – long-term instead of acute complications – does not appeal to me. Moreover, the clinical relevance and implications of the association found in this study will be hard to explain, as the relation between the acute perioperative response and the long-term state of the immune system is not being studied. With the data available, the only way to potentially do is would be to look at the association between preoperative and postoperative cell counts.

Dear Reviewer, thank you for your valuable comments. We shall explain our point of view:

The similar relation between inflammatory reactions activation in ACS patients undergoing PCI procedures [doi: 10.5114/aic.2020.95859. doi: 10.1177/03000605211010059. ]   and its' predictive role for survival was postulated.

The significance of inflammatory activation measured by CRP on cardiovascular outcomes was already presented [doi: 10.2174/1381612826666200717090334. ].

The detrimental effect of perioperative injury in surgical revascularization on survival is well known [ 10.1016/j.jtcvs.2007.12.029 , 10.1016/j.ejcts.2007.06.015. ].

We present the results of the study relating not to myocardial injury but inflammatory activation influencing the long-term survival.

We conclude that the inflammatory activation in the peri-operative period is not the cause of increased mortality rate by itself, but it rather indicates patients more prone for atherosclerosis progression. Inflammatory reactions are included in the origin of atherosclerosis, inflammatory exaggeration is a key component of ongoing atheroscleromatic process. Increased inflammatory reaction post-operatively shows predisposition for higher long-term mortality due to higher atheroscleromatic burden.

The anti-inflammatory therapies to modify pts outcomes with ischemic heart disease have been also proposed [doi: 10.1155/2021/5160728. and doi: 10.3389/fcvm.2021.726341

Patient selection: Why were patients presenting with acute coronary syndrome excluded? How long after the index surgery were patients follow-up for?

We aimed to collect the most possible homogeneity in the study group. The acute coronary syndrome is characterized by increased inflammatory status that may interfere the obtained results.

The mean follow up period was 4.7 +/- 1.3 years.

It is unclear in the methods section when in the perioperative period the cell count values were obtained – please provide more detail here. This is of key importance since the character of the acute perioperative response will vary over time. It is particularly confusing as table 1 contains preoperative cell count values, while further in the analysis ‘postoperative’ values are used.

The perioperative period was assigned as early reperfusion that occurs 12-24 hrs after surgical revascularization [doi: 10.1631/jzus.B1101010].

On several occasions in the methods actual results are mentioned. Please confine these to the results section.

We transferred some part of the methods section to the results.

The statistical analysis section is very unstructured – the order that the various steps are described in is not logical, which would make it impossible to repeat the analyses if data were provided. Also, transforming continuous variables into binary variables not only reduces statistical power significantly, but it also does not make a lot of sense in the context of the research question under study.

Dear Reviewer, thank you for you comments, the statistical section was improved by Prof. assoc. Michal Michalak from Statistic Department.

The three indices used are likely not independent, and it is therefore to be expected that there will be significant overlap between the various models presented. Correction for multiple testing should therefore likely be applied here.

The results of univariable and multivariable analysis were performed by Prof Michalak and the results of the analysis can be revealed on request. The model was constructed to avoid codependent variables.

It would be very useful if some more information were provided on residual confounding in this model. Given the very high HRs presented, it is hard to believe that there are no important either residual confounding or interactions that have not been accounted for in the modelling.

Dear Reviewer, the results of analysis performed by qualified statistician Prof M. Michalak from Department of Statistics. The data and performed analysis is available on request.

The way Table 2 is presented is misleading – displaying only significant results (of an uncorrected analysis) is not useful at all.

            Dear Reviewer, we corrected the Table 2 according to your valuable suggestion.

This manuscript is written in very poor English – both grammatically, as well as textual and typographic errors. Also, in various spot incorrect language is used (eg, L44: ‘cardiopulmonary bypass’ is incorrect. CA stands for Coronary Artery). It would need a significant revision before it can be considered again.

            Dear Reviewer, the manuscript was corrected by native speaker according to your request.

Reviewer 2 Report

The authors report a very interesting well written study. Usefull for scientific community

Author Response

Dear Reviewer 2,

I would like to thank you on behalf of all co-authors for your valuable suggestions.

I’m grateful you found the time and engagement to help us improve our manuscript.

We followed your suggestions as presented below.

          Kind regards

Tomasz Urbanowicz et. al.

Reviewer 2

The authors report a very interesting well written study. Usefull for scientific community

Dear Reviewer, thank you for your kind opinion.

Reviewer 3 Report

Tomasz et al., revealed the importance of inflammatory indices after coronary artery bypass surgery. This is an interesting study, even though the number of female patients (23% only). authors need to access inflammatory cytokines in their future research. 

Author Response

Dear Reviewer 3,

I would like to thank you on behalf of all co-authors for your valuable suggestions.

I’m grateful you found the time and engagement to help us improve our manuscript.

We followed your suggestions as presented below.

         Kind regards

Tomasz Urbanowicz et. al.

Reviewer 3

Tomasz et al., revealed the importance of inflammatory indices after coronary artery bypass surgery. This is an interesting study, even though the number of female patients (23% only). authors need to access inflammatory cytokines in their future research.

Dear Reviewer, thank you for your valuable suggestions. This is a retrospective study including consecutive pts referred for elective revascularization. In our material taken from clinical experience, the male to female ratio is 3:1.

We did focus on whole blood count parameters as possible simple predictive markers instead of advanced particles including cytokines. We believe that comparison between our results and specific inflammatory molecules in the prospective studies in necessary.